# A Teleophthalmology Support System Based on the Visibility of Retinal Elements Using the CNNs

**DOI:** 10.3390/s20102838

**Published:** 2020-05-16

**Authors:** Gustavo Calderon-Auza, Cesar Carrillo-Gomez, Mariko Nakano, Karina Toscano-Medina, Hector Perez-Meana, Ana Gonzalez-H. Leon, Hugo Quiroz-Mercado

**Affiliations:** 1Graduate Section, Instituto Politécnico Nacional, Mexico City 04440, Mexico; gus_auza@hotmail.com (G.C.-A.); crcg1995@hotmail.com (C.C.-G.); likatome@gmail.com (K.T.-M.); hmperezm@ipn.mx (H.P.-M.); 2Hospital Dr. Luis Sánchez-Bulnes, Asociación para Evitar la Ceguera in México, Mexico City 04030, Mexico; dranamghl@gmail.com (A.G.-H.L.); hugoquiroz@yahoo.com (H.Q.-M.)

**Keywords:** teleophthalmology, support system, quality assessment, deep learning, scanning laser ophthalmoscopes, convolutional neural networks, segmentation

## Abstract

This paper proposes a teleophthalmology support system in which we use algorithms of object detection and semantic segmentation, such as faster region-based CNN (FR-CNN) and SegNet, based on several CNN architectures such as: Vgg16, MobileNet, AlexNet, etc. These are used to segment and analyze the principal anatomical elements, such as optic disc (OD), region of interest (ROI) composed by the macular region, real retinal region, and vessels. Unlike the conventional retinal image quality assessment system, the proposed system provides some possible reasons about the low-quality image to support the operator of an ophthalmoscope and patient to acquire and transmit a better-quality image to central eye hospital for its diagnosis. The proposed system consists of four steps: OD detection, OD quality analysis, obstruction detection of the region of interest (ROI), and vessel segmentation. For the OD detection, artefacts and vessel segmentation, the FR-CNN and SegNet are used, while for the OD quality analysis, we use transfer learning. The proposed system provides accuracies of 0.93 for the OD detection, 0.86 for OD image quality, 1.0 for artefact detection, and 0.98 for vessel segmentation. As the global performance metric, the kappa-based agreement score between ophthalmologist and the proposed system is calculated, which is higher than the score between ophthalmologist and general practitioner.

## 1. Introduction

Eye diseases, such as diabetic retinopathy (DR), aged-related macular degeneration (AMD), glaucoma, retinopathy of prematurity (ROP), and so on, can be diagnosed by retinal image screening. Then, the patients with some diseases detected by the screening can receive an adequate treatment to avoid serious vision loss. Generally, persons who live in urban areas have more opportunities to receive this screening service in the eye hospital located in their community. However, persons who live in rural or remote areas do not have the same opportunities, due to a lack of infrastructures and specialists. Therefore, the vision loss of persons who live in the rural and remote areas cannot be prevented adequately [1,2]. Teleophthalmology can be considered as a promising solution to provide the screening service to persons who live in rural or remote communities [1,2].

The fundus cameras used in the screening can be classified into the mydriatic and non-mydriatic cameras. For mydriatic cameras, the eyedrop is required to dilate patient’s pupil, while for non-mydriatic camera, an eye-drop is not required. Considering that the eye-drop must be used under the supervision of an ophthalmologist, for a teleophthalmology framework under the absence of a specialist, the non-mydriatic fundus cameras are indispensable. Recently, the scanning laser ophthalmoscopes (SLOs) are considered as useful imaging instruments to take retinal images with a wide field of view (FOV). For example, the SLO fabricated by Optos Dytona provides a 200° FOV, which covers approximately 80% of the whole retinal region, contributing to a more reliable diagnostic of a broad range of eye diseases. When the 30–45° FOV fundus camera is used, it is necessary to take several photographs with different views to acquire a large region of the eye. While using a 200° FOV, just one central photograph covers the same region [3]. However, generally non-mydriatic cameras and SLOs produce more frequently low-quality images or even outlier images than mydriatic fundus cameras. Several documents report that approximately 5–15% of retinal images taken by mydriatic fundus cameras are useless for diagnostic due to its inadequate quality [4] and about 18% of retinal images taken by non-mydriatic cameras cannot be used for reliable diagnosis [5].

The low-quality images are produced by several causes, such as movement of patient, insufficient aperture of the eye causing obstacles due to eyelashes and eyelids, inadequate distance between patient and ophthalmoscope, inadequate camera focus, and inadequate luminance, etc. Specifically, in the case of SLOs, due to the mechanism of scanning for the ultra-wide FOV, several artefacts, such as eyelashes, eyelids, and even the nose, also appear in the retinal image. In the teleophthalmology framework, an inexpert person takes the retinal images and a low-quality and/or inadequate image may be transmitted to the central eye hospital. It will cause not only a waste of time, but also miss the detection of eye diseases of some patients who require timely treatments. Therefore, automatic retinal image quality assessment is very important for teleophthalmology.

Taking into account the above-mentioned situation, until now several automatic quality assessment methods for retinal images have been proposed, which are classified into three categories: generic featured-based methods [4,5,6,7], structural feature-based methods [8,9,10,11] and convolutional neural network (CNN)-based methods [12,13,14,15]. In the generic feature-based methods, some generic features of images, such as: sharpness, contrast, luminance and texture properties, are calculated and used to determine if an input image has good quality or inadequate quality for a reliable diagnosis [4,5,6,7]. In the structural feature-based methods, firstly some anatomical retinal structures, such as optic disc (OD), vessel tree and macula, are localized and/or segmented. Then the visibility or clarity of these anatomical structures are evaluated to determine the quality of retinal image [8,9,10,11]. The performance of the structural feature-based methods depends strongly on a precise segmentation or localization of each anatomical element mentioned above. However, because these methods are similar with the human evaluator, the methods have been gaining higher support among ophthalmologists than the generic feature-based methods. In the CNN-based methods, using the powerful classification capabilities of deep neural networks together with deep learning algorithm, a quality classification of an input retinal image is performed [12,13,14,15]. Almost all methods provide classification of an input image into two classes: good quality image and inadequate quality image, or more than three classes, considering intermediate classes. However, these methods cannot provide any meaningful support for the patient and the operator of the ophthalmoscope to obtain a better-quality retinal image, because the reasons of the poor quality are not provided to them. 

In this paper, we propose a support system for teleophthalmology framework based on non-mydriatic fundus cameras, basically on the scanning laser ophthalmoscopes (SLOs). In the proposed system, some possible reasons of low-quality image are provided, if the captured image is determined as inadequate for a reliable diagnostic. According to these reasons, the operator of the SLO can recapture the image, trying to acquire a better-quality image. The retinal images with some specific eye disease may be determined to have inadequate quality by the proposed system because important anatomical structures, such as OD, are not present in the image. Considering this situation, we decide that the maximum repeat number of captures is set to five, following an advice of ophthalmologists. When the captured image is assessed as good quality by the proposed system, the retinal image together with several useful information, such as the OD image and binary vessel image, are transmitted to the central eye hospital. The ophthalmologist can do a reliable diagnosis using the information under the absence of the patient.

The proposed support system is composed by four steps. The first step is the CNN-based OD segmentation and in the second step the quality of the detected OD region is analyzed. In the third step, the region of interest (ROI), which is composed by the OD, the superior and inferior arcades of principal vessels and macula, is segmented and the grade of obstruction of the ROI by artefacts is evaluated using the CNN-based real retinal area segmentation. In the real retinal area segmentation, super-pixel technique is used to generate the ground truth (GT) for the training, which is based on the method proposed by [16]. However, instead of using standard classification methods as [16], we used a CNN architecture to perform this task. If the grade of obstruction is larger than a predetermined threshold, the proposed system considers the image as inadequate due to the obstruction by artefacts. If an acquired retinal image passes through all three steps, the acquired image is considered as good quality, then in the fourth step, the vessels of the ROI are segmented by a CNN-Based segmentation method. Finally, the acquired good-quality retinal image together with the segmented OD image, a binary vessel image, and a report generated by the system are transmitted to the central eye hospital. If an acquired image cannot pass some of the firsts three steps after the maximum number of repetitions (five times), the system transmits the best-quality image among all five acquired images, together with a generated report, to the central eye hospital. In this manner, the proposed support system not only helps the operator and the patient to acquire a better-quality retinal image, but also provides several useful information to the ophthalmologist in the central eye hospital. The ophthalmologist in the central hospital can take more reliable diagnostic using all information about the image. Although the proposed system is developed for the SLO, the system can be used for any teleophthalmology system based on conventional fundus cameras.

The rest of the paper is organized as follows: in Section 2, we provide some analyses of retinal image quality and criteria of quality assessment used to construct the proposed system, and the detail of the proposed system is mentioned in Section 3. We provide partial and global performances of the proposed system, as well as some system output in Section 4. Finally in Section 5, we conclude this work.

## 2. Criteria for Retinal Image Quality Assessment

Unlike the quality assessment of natural images, the retinal image quality assessment (RIQA) strongly depends on the possibility of reliable diagnostic from the images by ophthalmologists. Therefore, first we must establish the criteria for the RIQA. In this section, we analyze a large number of low-quality images that impedes reliable diagnostics. There are several factors for low-quality retinal images, and in many cases some factors are combined. Although there are several public-domain retinal image datasets, whose purposes are different from the quality assessment. The principal purposes of these public-domain dataset are the evaluation of automatic eye disease detection systems, such as the automatic DR grading system and the automatic glaucoma detection system. Then, the retinal images of these datasets generally present good quality for their diagnostics. Therefore, we constructed our own dataset with retinal images during last three years. The management of the image dataset is performed according to the guidelines of the Declaration of Helsinki [17]. The image acquisition equipment used is a Daytona SLO with 200° FOV fabricated by Optos. The images are color images acquired by red and green laser beams with the size 3900 × 3072 pixels, which are stored in TIFF or JPEG format. We collected 1288 images in total.

Figure 1 shows several low-quality images taken by this SLO, as well as some high-quality images that are able to use for reliable diagnostics. Figure 1a,b shows outlier images caused by an inadequate distance between ophthalmoscope and patient’s eye or an inadequate movement of the patient at the moment of the image capture. Figure 1c,d show the blurred images caused by a probable movement of the eye or a maladjustment of the equipment. Figure 1e,f show images with obstruction by eyelashes and eyelid. Finally Figure 1g,h shows good-quality images which can be used for reliable diagnostics.

After analysis of low-quality images produced by several factors, we conclude some relationships among the visibility of the anatomical elements and image quality, which can be summarized as follows: (1) All of the outlier images do not contain an optic disc (OD). The OD shows as a high intensity spot, however it can be distinguish from other high intensity spots caused by some white lesions, because the OD shows some particular features due to the presence of vessels inside of it. (2) The blurriness of the OD reflects directly the blurriness of the whole retinal image, then if the region of the OD is blurred, the quality of whole images impedes reliable diagnostics. (3) If some artefacts, which are principally patient’s eyelash and/or eyelid, obstruct the ROI, it is difficult to observe important lesions in the ROI, although the detected OD is clear.

## 3. Proposed Teleophthalmology Support System

Taking into account the above-mentioned analysis about the retinal image quality, we propose a teleophthalmology support system, which is composed of mainly four steps: OD region segmentation, blurriness analysis of the OD region, ROI segmentation and computation of the grade of obstruction by the artefacts, as well as vessels segmentation in the ROI. The last process is performed only when the input image passed the earlier three steps and is considered as a good-quality image. The block diagram of the proposed system is shown in Figure 2.

A retinal image acquired by a SLO or a fundus camera is introduced to the OD detection process, in which a box-like region (bounding box) centered by the OD-center is detected. If the OD is not detected by the system, then the input image can be considered as outlier or it contains some disease in which the OD does not appear. In this case, the operator receives the report-1 with some probable reasons, and the image acquisition must be repeated. If the OD is detected, then the image quality of the detected OD region is analyzed as second step. In this step, if the OD region is classified as blurred, the operator receives the report-2 with some probable reasons of this result and the image acquisition must be repeated. If the OD region is sufficiently clear, then the ROI is segmented taking account of the estimated center and the radio of the OD. The macular region is also estimated in this step. In some diseases, such as the DR, the AMD and so on, the ROI is especially important, therefore any obstruction of this region makes difficult a reliable diagnostic. Therefore, in the third step, the obstruction rate by artefact is calculated using the real retinal region segmentation using super-pixels. If obstruction rate is larger than the predetermined threshold (Th_OB_), then the operator receives the report-3 related to the obstruction by the artefact. Finally, if the input image passed through the three steps, it can be considered as good-quality image, then the vessel of the ROI is segmented to generate a binary vessel image, which is transmitted to the central hospital together with the whole input image and the OD image. As mentioned before, the maximum number of repetitions (Th_COUNT_) is set to five, according the consensus of the ophthalmologists. In the best case of the proposed system, the first captured image passes through the three evaluations and finishes the process. While in the worst case, five retinal images must be taken, in which the retinal image with the largest QL value, together with additional information are transmitted to the central eye hospital. The Table 1 indicates the contents of each one of the three reports. The details of the four steps, including the vessel segmentation step, are mentioned in the consecutive subsections. 

### 3.1. Optic Disc (OD) Detection

For the object detection, several CNN architectures have been proposed in the literature. The single shot detector (SSD) [18], region-based CNN (R-CNN) [19] and faster R-CNN (FR-CNN) [20], are the principal architectures for object detection. According to the previous research [21], the detection performance of the FR-CNN is superior to the SSD, when the size of the interest object is small. Our interest object is an OD and it is exceedingly small, occupying only 0.2% of the whole retinal image taken by the SLO, as shown in Figure 4. Therefore, we selected FR-CNN as OD detection CNN. The FR-CNN is composed of a feature extraction network based on some off-the-shelf CNN architecture, such as Vgg16, AlexNet and MobileNet, followed by two sub-networks, as shown in Figure 3. The first sub-network is a region proposal network (RPN), which is trained to generate object proposals regions (object or background). The second sub-network is trained to predict the actual class of each proposal region. In our proposed system, the interest object is an OD, therefore the second sub-network provides the probability about the detected region, if it contains a real OD then the probability Pr(OD) is high, otherwise it is low. To determine if the segmented region is the OD or not, we compare Pr(OD) with a threshold ($Th_{OD})$. If $\mathrm{Pr}\left(OD\right)>Th_{OD}$ then the detected bounding-box is considered as OD region. To train FR-CNN, first the retinal images are resized according to the CNN architecture used in the feature extraction network, thus if the Vgg16 is used in this first step, the size of the input image is 224 × 224 × 3. To generate the training dataset, the bounding-box is manually labeled in each training image of the dataset, as shown in Figure 4. The size of bounding-box varies slightly depending on the image. The mean height and width are 167.85 and 169.91 pixels, and their variance are 23.93 and 26.09 pixels, respectively. After data augmentation is applied to the 1288 original images, we obtained 12,880 images in total. The 70% of them are used for training and the 30% remaining for evaluation. In the training process, we use the Stochastic Gradient Descent with Momentum (SGDM) as optimizer under the following hyper-parameters: the momentum of 0.9, a learning rate of 0.0001, batch size is one image and the number of epochs is 20. We used transfer learning for the feature extraction network to reduce the training time.

### 3.2. Quality Analysis of the OD Region

To analyze the quality of the OD, we also use transfer learning which is applied to the off-the-shelf CNNs, such as AlexNet and Vgg16 [22,23], because our dataset is relatively small, and transfer learning is usually much faster and easier than training a network with randomly initialized weights from scratch. Firstly, all images are resized to input image size, according to the selected CNN for this task, for example in AlexNet, input image size must be 227 × 227 × 3. The dataset is generated from 11,733 images which contain the OD after the OD detection process is applied to the all 12880 images. The images containing the OD are divided into 70% for training and the remaining 30% for testing. With transfer learning, after trial and error, we found a configuration with the best performance for almost all CNN architectures, in which the Conv layers are frozen up to Conv2-2, and the other layers are retrained, including the fully connected layers, as shown in Figure 5 (in the case of Vgg16). For fine tuning, we used the SGDM method as an optimizer under the following parameters: the momentum is set to 0.9, the learning rate to 0.0001 with batches of 64 images during 30 epochs as well as other CNNs. In the evaluation, the Softmax result, which is the good-quality probability, is compared with the predefined threshold value $Th_{OQ}$ to determine the quality of the input OD image. Figure 6 shows some examples of the ODs with good quality Figure 6a and bad quality Figure 6b.

### 3.3. Obstruction Analysis in the ROI

Based on the UK National Screening Committee Guideline [24], we define the region composed by the OD, macular region, upper and lower arches of the principal vessels as the region of interest (ROI) as shown in Figure 7. Once we have obtained the bounding-box of the OD, which is represented as coordinate (X, Y) of its upper-left point and its side length, it is easy to estimate the ROI in the input retinal image. The diameter of the OD, denoted as DD, is approximated as the side length of the bounding-box, and considering the eye (left or right), the location of the ROI is estimated as follows. The upper-left point of the ROI for the left eye $\left(x_{L0},y_{0}\right)$ and for the right eye $\left(x_{R0},y_{0}\right)$, height ($l_{1}$) and width ($l_{2}$) of the ROI are calculated by
(1)$$x_{L0}=X-0.5\cdot DD\;x_{R0}=X-4.0\cdot DD$$
(2)$$y_{0}=Y-2.5\cdot DD$$
(3)$$l_{1}=6.0\cdot DD$$
(4)$$l_{2}=5.5\cdot DD$$

Furthermore, we can extract the macular region from the ROI which is the most important part for diagnostics of almost all eye diseases. The parameters of macular region are given by
(5)$$M_{Lx}=X+2\cdot DD\;M_{Rx}=X-3\cdot DD$$
(6)$$M_{y}=Y-0.5\cdot DD$$
(7)$$l_{m1}=2.5\cdot DD$$
(8)$$l_{m2}=2\cdot DD$$
where $M_{Lx}$ and $M_{y}$ are the upper-left coordinates of the macular region of the left eye, while $M_{Rx}$ and $M_{y}$ are the upper-left coordinates of macular region of the right eye. $l_{m1}$ and $l_{m2}$ represent the height and the width of macular region. Figure 7 shows the ROI and macular region of both eyes.

To effectively detect some artefacts, such as eyelashes and eyelid, we use the super-pixel technique [25], because eyelashes and eyelid, respectively, present the same or similar color. For example, eyelashes present a similar dark color, while eyelid presents a similar skin-color. We set the number of super-pixels for the whole image equal to 1000. This number of super-pixels is suitable to group correctly pixels belong to eyelashes as well as pixels belong to eyelid. An example of this process is shown in Figure 8, in which Figure 8a shows the original ROI, while Figure 8b shows the ROI represented by super-pixels. 

Using images represented by super-pixels, we can generate GT (Ground Truth) effectively to train the SegNet CNN [26] to segment real retinal area from artefacts. Figure 9 shows an example of the generated GT, in which the real retinal area is indicated by a white color (a), and the original image (b) is also shown.

The SegNet [26] has a similar architecture to the U-Net [27], however the up-sampling process of the SegNet employs a technique called unpooling, which saves the locations of a maximum element when the max-pooling operations are performed in the encoder process. In this manner, the SegNet can perform the decoding process more efficiently than the U-Net. Figure 10 shows the SegNet architecture used for this task.

To generate the training dataset for this task, we selected randomly 100 patches with size 224 × 224 × 3 from each one of 10 images with artefacts. Then data augmentation technique is applied to these 100 patches to increase artificially to 500 patches per image. In total, we generated 5000 patches for train the SegNet. The data augmentation techniques used for this task are random rotation within [−60°,60°], vertical and horizontal random reflections and random translation with 5 pixels in horizontal and vertical directions are performed. We used 80% of the dataset for training and the remaining 20% for test. In the training process, ADAM is used as optimizer under a constant learning rate of 0.0005, a batch size is equal to 8 and 400 epochs were required to converge.

Using the trained SegNet, the retinal image that passed the previous two evaluations (see Figure 2) is segmented into real retinal area and artefacts. The percentage $\left(P_{OB}\right)$ of the pixels belonging to the artefact in the ROI is calculated, and if it is larger than a predetermined threshold value $Th_{OB}$, then we consider that the ROI is highly obstructed by artefacts, and the image acquisition must be repeated. This situation is notified by the report 3.

### 3.4. Vessel Segmentation 

For the vessel segmentation, the SegNet shown by Figure 10 is also used. To train the SegNet, vessels are segmented manually from ten retinal images and the data augmentation technique was applied to the patches of size 256 × 256 × 3 obtained from these ten images. In the data augmentation, the same geometric transformations used for obstruction analysis, such as random rotations, random reflection, and random translation, are performed. For the vessel segmentation, in total, we generated 10,000 patches from ten images. Of them, 90% of the generated patches are used for training and the remaining 10% for validation. It is important to mention that due to the imbalanced problem, where the number of background pixels is much larger than that of vessel pixels, the network must be trained several times to get a better segmentation performance. In the training of SegNet, the SGDM optimizer was used, under the learning rate 0.0005, the momentum 0.9, the mini-batch size 32, and the epoch number 100. 

Table 2 resumes the architecture, optimizer and hyper-parameters used in each process of the proposed system. The values of the hyper-parameters are determined by trial and error.

## 4. Experimental Results

In this section, we first evaluate the performance of each individual process that composes the proposed teleophthalmology support system, which are the OD detection, the OD quality analysis, obstruction analysis in the ROI, and vessel segmentation. The metrics used for performance evaluation are accuracy, sensitivity, and specificity, which are given by
(9)$$Accuracy=\frac{TP+TN}{TP+FN+FP+TN}$$
(10)$$Sensitivity=\frac{TP}{TP+FN}$$
(11)$$Specificity=\frac{TN}{TN+FP}$$

Considering that, in teleophthalmology, a short processing time is important, the processing time is also evaluated.

Finally, the global performance of the proposed system is evaluated, showing the agreement grade between experts of the eye hospital and the proposed system, which is calculated by Cohen’s kappa index. In Section 4.6, we provide some examples of the system output when several retinal images are introduced to the proposed support system. As mentioned before, the system output is composed of a retinal image with the best quality among all trials (in worst case, five times), a report which is one of the three reports given in Table 1, OD image, the ROI image, the macular image and the segmented vessel image.

### 4.1. OD Detection 

As mentioned in Section 3.1, we selected the FR-CNN architecture with a CNN as the feature extraction network for this task. To select a best CNN for our purpose, we evaluate several off-the-shelf CNNs, such as Vgg16, AlexNet and MobileNet as the feature extraction network of the FR-CNN. Table 3 shows the performance comparison of the OD detection among four different methods, which are the FR-CNN with Vgg16, the FR-CNN with AlexNet, FR-CNN with MobileNet and histogram matching method [28], using our own dataset generated by the Opto’s SLO, while Table 4 shows the performance comparison using the public dataset DRIMBD [10]. In the proposed methods (FR-CNN with Vgg16 and FR-CNN with AlexNet), the threshold value Th_OD_ is set to 0.9. In both tables, the average processing time requiered per image for each method is added. The histogram matching method is one of the most used hand-crafted OD localization method, we evaluate Dehghani’s method [28] for comparison purpose, in which a template of histogram for each color channel of the OD region is constructed using several retinal images, and then a correlation between histograms (three channels) of each candidate region and templates is calculated to select a region with the highest correlation value as the OD region.

From Table 3 and Table 4, we conclude that the FR-CNN with AlexNet as the feature extraction network provides a better performance compared with other combination and the histogram matching method. The reason of the low specificity values in all methods for our own dataset, as shown by Table 3, is that the dataset contains several retinal images with severe uncommon eye diseases that do not present the OD or the ODs with very different appearance from the normal ODs. The histogram matching method provides good performance for good-quality images [28], in which the characteristic variation between the OD template and input ODs is small. However, our own dataset and the DRIMBD dataset contain several low-quality images, in which the characteristic variation between the OD template and input ODs is considerably large. We consider that this is the principal reason of low detection performance of [28] using our own dataset and DRIMBD. Figure 11 shows some examples of the OD detection, where the value in the upper part of the detected region is the probability.

### 4.2. Quality Analysis of the OD Region

In the quality analysis of the OD region, we first create a new dataset consisting of 11733 OD regions. To this end we applied our OD detection method to both: our own dataset composed of 12,880 images and the public database DRIMBR [10]. The created OD dataset is divided as good quality (6577 images) or bad quality (5156 images), according to the opinions of ophthalmologists. To select a best method for the quality analysis of the OD region, we evaluated the performance of several CNN-based methods, such as AlexNet, Vgg16, and some hand-crafted feature based methods, such as Cumulative Probability of Blur Detection (CPBD) [29], Blur Metric [30] and the combination of CPBD [29] and Blur Metric [30]. As mentioned before, in the CNN-based methods, we used transfer learning to train the networks, in which 70% of the OD dataset are used in the training, and the rest is used for test. The threshold value Th_OQ_ for two CNN-based methods is set to 0.95. Table 5 shows the performance comparison among two CNN-based methods and three hand-crafted feature based methods, together with the average processing time for each OD region. 

From the table, we can conclude that the CNN-based method with Vgg16 shows the best performance with 0.03 second of the average processing time by each OD, therefore we selected Vgg16-based quality analysis for the proposed support system. Figure 12 shows an example of classification of the ODs with good-quality (a) and inadequate quality (b).

### 4.3. Obstruction Analysis in the ROI

To determine if the obstruction by artefacts in the ROI impedes a reliable diagnostic or not, we set the threshold value Th_OB_ equal to 0.1 (10%), which is the tolerable rate of obstruction in the ROI by artefact. Figure 13 shows some segmented ROI with eyelashes and results of artefact detection, Figure 13a,c are original ROIs and Figure 13b,d shows the results of segmented artefacts, in which Figure 13a,c present, respectively, the artefact rates $P_{OB}$ 3.77% and 41.85%. Then, the ROI shown by Figure 13a is considered as an adequate ROI, while Figure 13c is considered as an obstructed ROI by artefacts due to its obstruction rate becomes larger than the threshold value Th_OB_ (10%). Although there are few images with high obstruction (only 5% of the test set), all of them are detected correctly as obstruction by artefacts, providing an accuracy of 1.0. If the obstruction is detected, the report-3 is provided to the operator.

### 4.4. Vessel Segmentation in the ROI

To select the best component for SegNet, we evaluated several CNN architectures, such as Vgg16 [23] and MobileNet [31]. The performance comparison using accuracy, sensibility, specificity, together with the processing time using our own dataset are shown in Table 6. There are several methods for the vessel segmentation, such as multiscale approaches [32,33], the deformable model [34], and CNN-based method [35]. Table 7 shows the performance comparison using the DRIVE dataset, which is a public dataset for vessel segmentation [36]. From Table 6 and Table 7, we conclude that the SegNet with Vgg16 shows a better performance, especially, the sensibility of the proposed segmentation is higher than other methods, guaranteeing the detection of small and thin vessels. We evaluate the segmentation performance of our proposed method, the SegNet with Vgg16, with the DICE and the Jaccard metrics, obtaining 0.979 with the DICE metric and 0.989 with the Jaccard metric. Figure 14 shows some vessel segmentation result together with the segmented vessels superimposed on the original retinal image. We show the vessels of whole images for their better observation. 

### 4.5. Agreement Grade Between Expert and the Proposed System

To evaluate the global performance of the proposed teleophthalmology support system, we calculate the Cohen’s kappa index (κ), which indicates the agreement grade between two graders, which is expressed by (12).
(12)$$\kappa =\frac{p_{o}-p_{e}}{1-p_{e}}$$
where $p_{o}$ is relative observed agreement among evaluators and $p_{e}$ is hypothetical probability of chance agreement, which are given by
(13)$$p_{o}=\frac{Ngg+Nbb}{T}$$
(14)$$p_{e}=p_{Yes}+p_{No}$$
where
(15)$$p_{Yes}=\frac{Ngg+Ngb}{T}\times \frac{Ngg+Nbg}{T}\;p_{No}=\frac{Nbg+Nbb}{T}\times \frac{Ngb+Nbb}{T}$$
where *T* is total number of images used for evaluation, and the value Ngg is the number of images that both evaluators consider as good quality for reliable diagnosis and Nbb is the number of images that both evaluators consider as inadequate quality. The values of Ngb and Nbg indicate the number of images in which the opinions of both evaluators do not agree.

In this experiment, two expert ophthalmologists and five general practitioners participated. Two expert ophthalmologists are working in the retinal department of the eye hospital and they review retinal images as their daily works. Five general practitioners who graduated recently from medical school, but have not finished yet the ophthalmology specialty. To evaluate systematically the performance of the proposed system, we generated several retinal images with different blurriness, different luminance and different levels of contrast. These generated images are then presented to the two experts (E1 and E2) and the five general practitioners (G1, …, G5), asking if the presented image can be useful for its diagnostic or not. Table 8 shows the parameters used to generate retinal images and Figure 15 shows some distorted retinal images generated synthetically. In total, we generated 320 images (T = 320) with different distortions (10 images with 32 variation per image).

Table 9 shows the results of the Cohen’s kappa index between two experts (E1 and E2), the average kappa index value $Av\left(\kappa \left(E_{i},G_{j}\right),i=1,2,\;j=1,\;\ldots ,5\right)$ of two experts and five general practitioners (G1,G2,…,G5) and the average value $Av\left(\kappa \left(E_{i},P\right),i=1,2\right)$ of two experts and the proposed system (P). These average values are calculated as
(16)$$Av\left(\kappa \left(E_{i},G_{j}\right),i=1,2,\;j=1\ldots 5\right)=\frac{1}{10}\overset{2}{\underset{i=1}{\sum }}\overset{5}{\underset{j=1}{\sum }}\kappa \left(E_{i},G_{j}\right)\;Av\left(\kappa \left(E_{i},P\right),i=1,2\right)=\frac{1}{2}\overset{2}{\underset{i=1}{\sum }}\kappa \left(E_{i},P\right)$$
where κ(A,B) is Cohen’s kappa index between A and B.

We can observe from the Table 9, that the proposed system provides a higher agreement grade respect to the experts, compared with the agreement between experts and general practitioners.

### 4.6. Examples of the System Performance and Overall Processing Time

In this section, the proposed teleophthalmology support system framework together with some examples of the proposed system when some retinal images captured by the SLO are introduced. In the proposed framework, the captured retinal image with the best quality is shown in the “Selected Retinal Image” section, the OD image is shown in the “Optic Disc” section. In the “Quality OD” section indicates the quality of the OD obtained by the system, the presence of the obstruction in the ROI is also reported in “Artefact in ROI” section, the side of the eye is reported in the section “Eye” as “Left” and “Right”. The supplemental images such as image of the ROI, image of the macular region as well as segmented vessel image, which is a binary image, are shown in the respective section in the framework. Figure 16 shows an output example of the support system when an input retinal image is of good quality, by passing all quality evaluations in the proposed support system, while the Figure 17 shows another example of the system output, where a low quality retinal image is introduced in the proposed system. For more examples, please visit the Appendix A. 

We measured the processing times of each case for the proposed ophthalmology support system. The processing time is measured in the Intel i7-8700^®^ at 3.2 GHz, NVIDIA GeForce GTX 1080 Ti^®^, and 32 GB of RAM. The Table 10 shows the average elapse times. It is worth noting that these processing times include images, report generation and display times. From Table 10, if the input image is high quality, it takes approximately 11.3 s.

## 5. Conclusions

In this paper, we proposed a teleophthalmology support system, in which the quality of the input retinal images taken by a non-mydriatic ophthalmoscope, such as scanning laser ophthalmoscopes (SLO) and a fundus camera, are automatically evaluated. The proposed support system not only evaluates the quality of the retinal image, but also provides reasons why the input retinal image presents inadequate quality for its diagnosis to the operator of the ophthalmoscope as well as to the patient. In this way, the image acquisition process can be repeated under the conscience of the operator and patient, guaranteeing the transmission of the higher-quality retinal image to the central eye hospital.

To obtain these useful properties in the teleophthalmology framework, several anatomical elements, such as optic disc (OD), vessels and macula are segmented. These elements are then analyzed to determine their visibilities and clarities using the several CNN architectures according to the criteria recommended by the UK National Screening Committee [24]. Additionally, the retinal image is segmented into the real retinal area and artefacts, composed mainly by eyelashes and eyelid, using the SegNet CNN. Here, if the ROI is highly obstructed by artefacts, the proposed support system notifies to the operator, as well as the patient, to repeat the acquisition process. The patient can use his/her finger to open his/her eye if necessary. The maximum number of repetitions is set to five, according to the suggestion of the expert ophthalmologists. 

We evaluated the performance of each process of the proposed support system, which are compared with some hand-crafted based methods and other CNN architectures. The global performance is evaluated using Cohen’s kappa index compared with the agreement index among expert ophthalmologists and general practitioners. The accuracies of the OD detection, the OD quality assessment, artefact detection and vessel segmentation are 0.93, 0.86, 1.0, and 0.98, respectively. The kappa index between the proposed system and experts is 0.6146, while this value between expert and the general practitioners is 0.5404, concluding that the global quality evaluation performance of the proposed support system is better than the performance of the general practitioners. 

The proposed support system provides a best-quality image among all acquired images (maximum five times in worst case). If the input image is considered good-quality by the system, then its OD image, ROI image, including macular image, and the segmented binary vessel image are provided and transmitted to the central eye hospital together with the input image. If the input image cannot pass the three quality evaluations after the maximum number of repetitions, then the proposed system provides only the better-quality image together with a report generated by the system, which are also transmitted to the central eye hospital for further analysis. In this manner, the proposed system not only provides a useful information to the operator of ophthalmoscope and the patient, but also to the ophthalmologist in the central hospital. Using all information about the retinal image, an ophthalmologist can provide a reliable diagnosis in the absence of the patient. Some captured screens of the proposed systems show its functionality.

The proposed system is considered as a part of our teleophthalmology project, in which all required computation will be carried out in the cloud environment with appropriate computer power and memory size. As a future work, we are working on an automatic diagnostic system, such as the automatic DR screening system. 

## Figures and Tables

**Figure 1 sensors-20-02838-f001:**
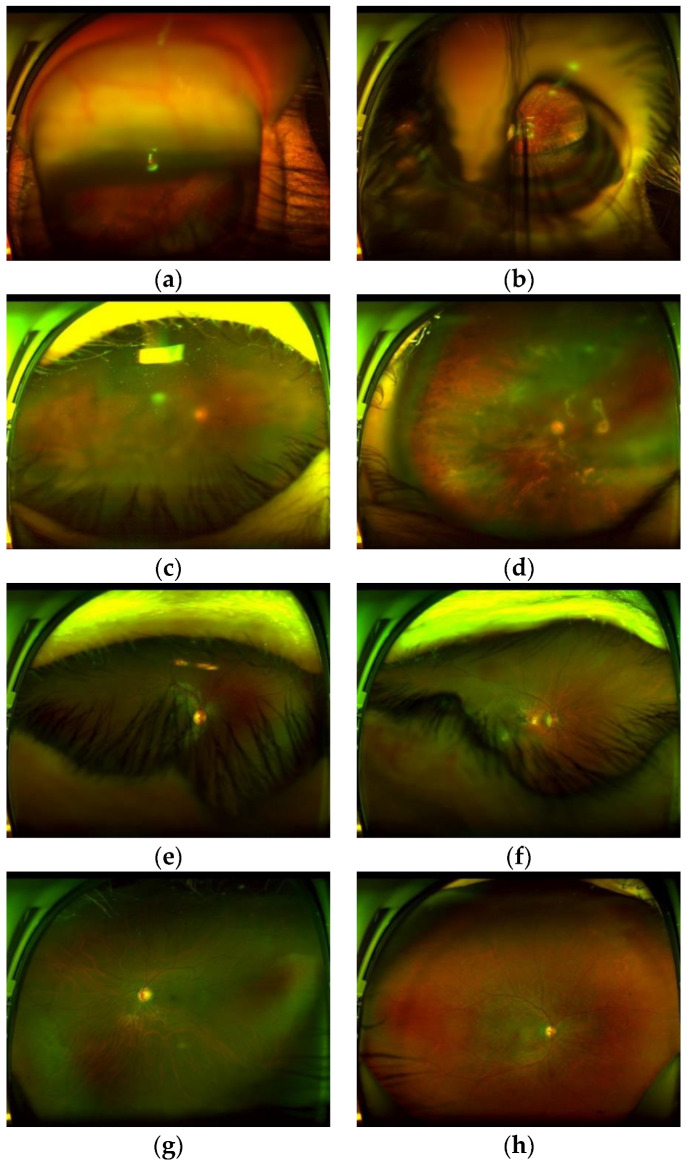
Several low-quality images and high-quality images taken by the SLO. (**a**,**b**) outlier images, (**c**,**d**) blurred images, (**e**,**f**) images with obstruction by eyelash, (**g**,**h**) high-quality images.

**Figure 2 sensors-20-02838-f002:**
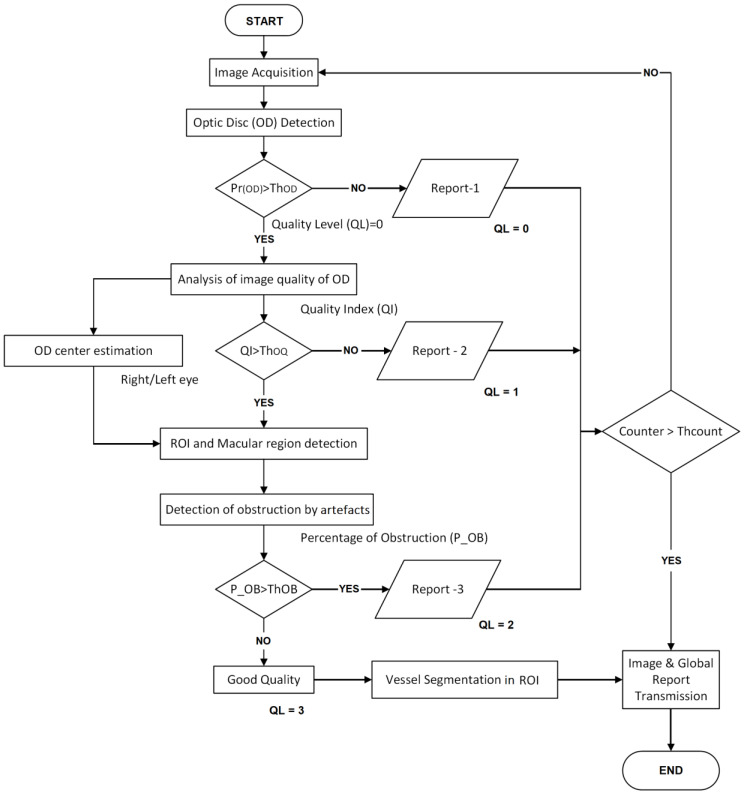
Block diagram of the proposed teleophthalmology support system.

**Figure 3 sensors-20-02838-f003:**
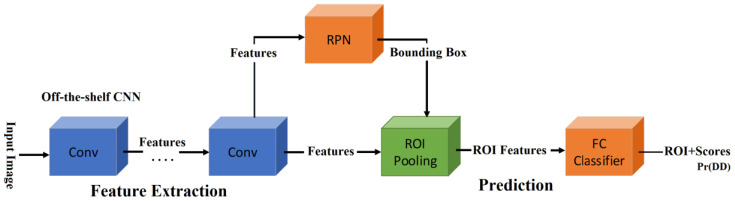
Faster Region-CNN (FR-CNN) architecture [20].

**Figure 4 sensors-20-02838-f004:**
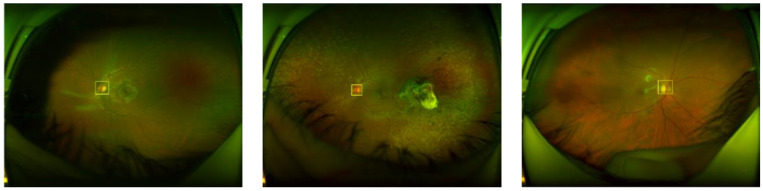
Examples of manual bounding-box labeling of the OD.

**Figure 5 sensors-20-02838-f005:**
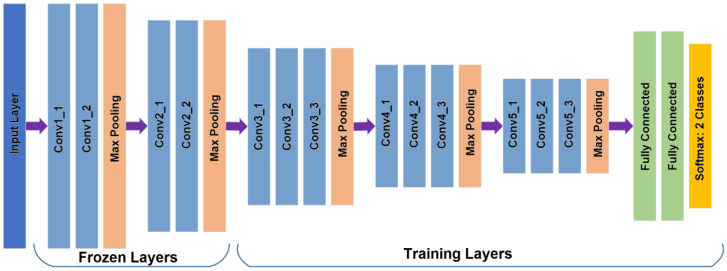
The Vgg16 CNN architecture [23] with different final layer to classify two categories showing the frozen and training layers.

**Figure 6 sensors-20-02838-f006:**
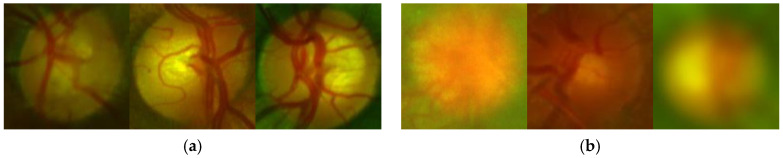
Examples of the ODs with good quality (**a**) and inadequate quality (**b**).

**Figure 7 sensors-20-02838-f007:**
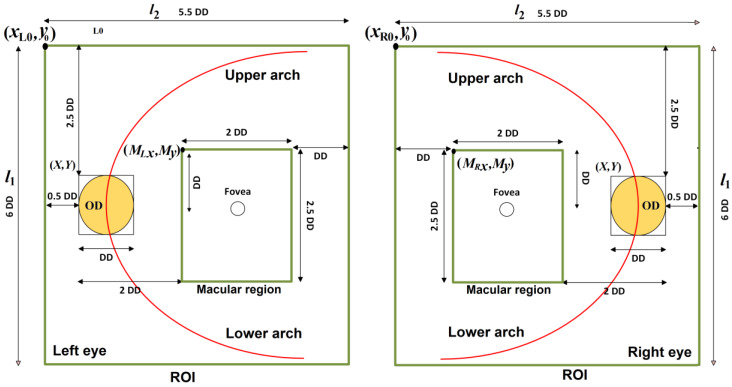
ROI and macular region of the left and right eyes.

**Figure 8 sensors-20-02838-f008:**
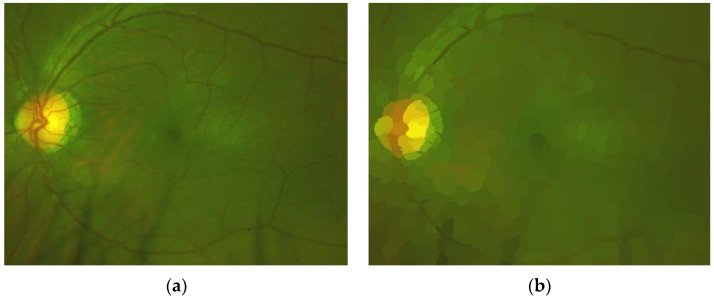
(**a**) Original ROI and (**b**) ROI represented by super-pixels.

**Figure 9 sensors-20-02838-f009:**
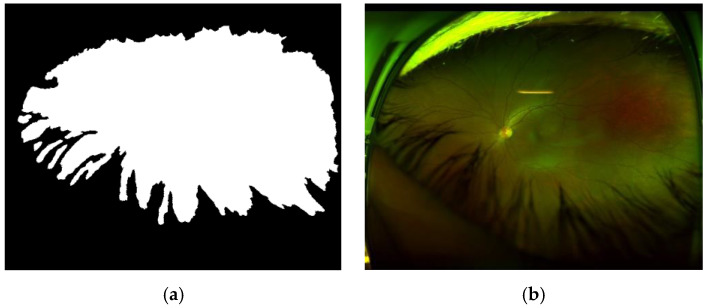
(**a**) GT with white pixel presented real retinal area and (**b**) the original image.

**Figure 10 sensors-20-02838-f010:**
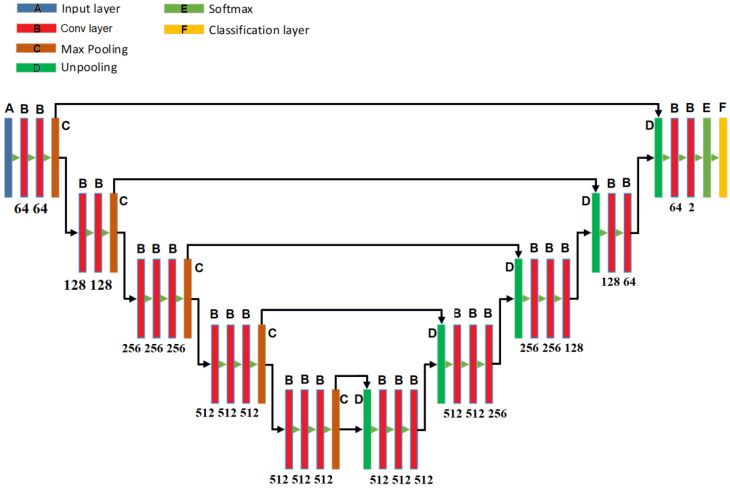
The architecture of SegNet used in the proposed system.

**Figure 11 sensors-20-02838-f011:**
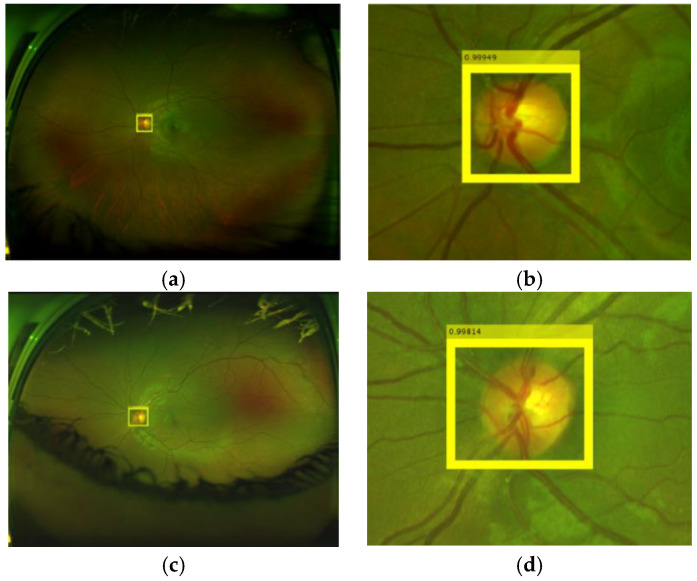
Examples of OD detections. (**a**,**b**) OD detection in the input image, (**b**,**d**) are amplified version of (**a**,**c**), respectively.

**Figure 12 sensors-20-02838-f012:**
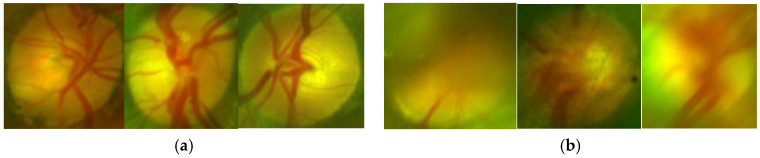
Examples of classification of quality of the OD. (**a**) ODs classified as good-quality, (**b**) ODs classified as bad-quality.

**Figure 13 sensors-20-02838-f013:**
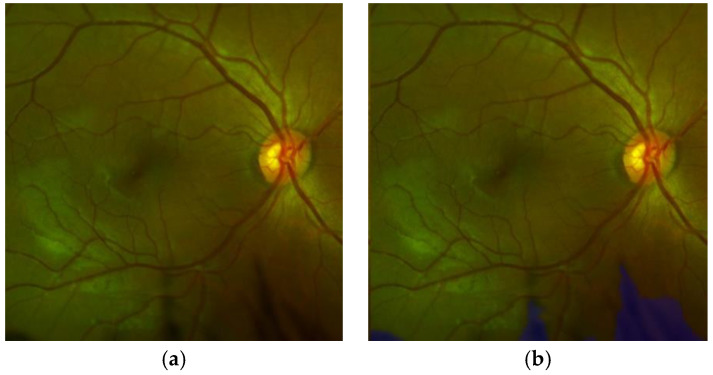
Examples of artefact segmentation in the ROI. (**a**,**c**) images of the ROI and (**b**,**d**) the ROIs with artefact detected.

**Figure 14 sensors-20-02838-f014:**
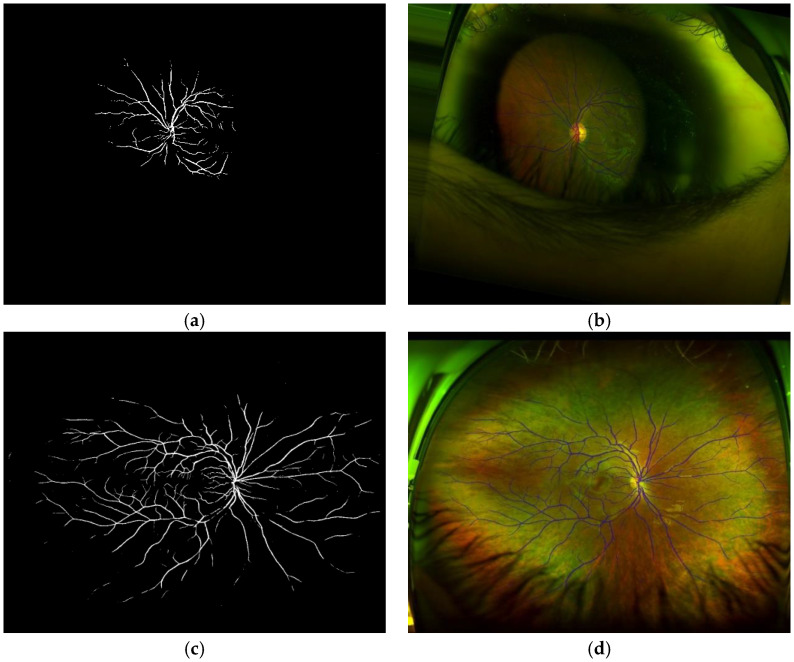
(**a**,**c**) results of vessel segmentation. (**b**,**d**) the segmented vessel superimposed on the original images.

**Figure 15 sensors-20-02838-f015:**
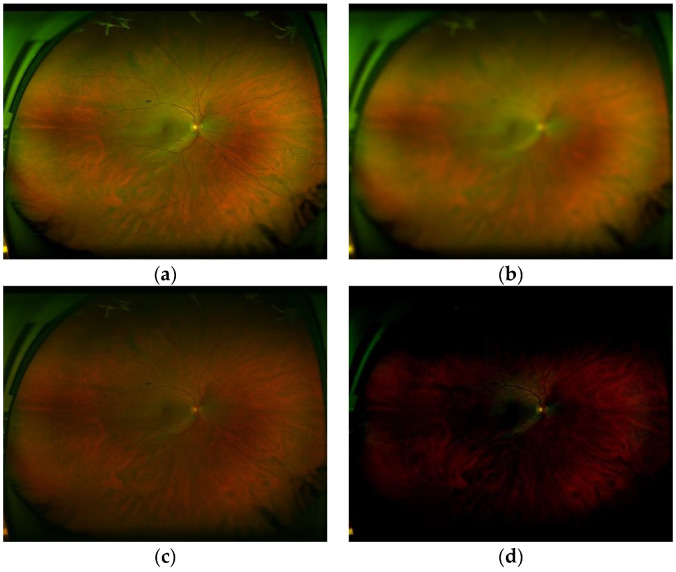
(**a**) good-quality images (**b**) inadequate quality due to blurriness, (**c**) inadequate quality due to low contrast and (**d**) inadequate quality due to inappropriate luminance.

**Figure 16 sensors-20-02838-f016:**
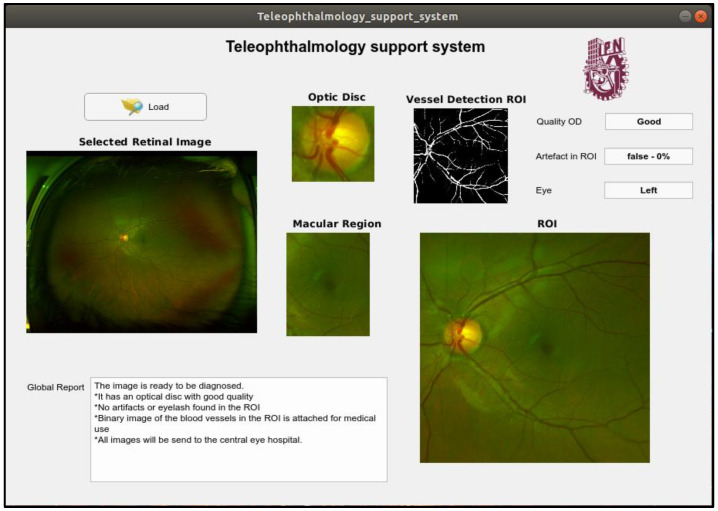
An example of output of the support system when the input image is good quality.

**Figure 17 sensors-20-02838-f017:**
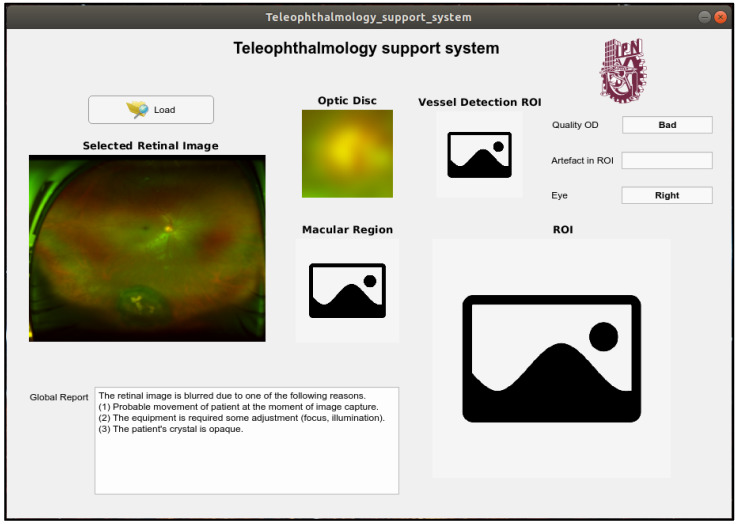
An example of output of the support system when the input image is inadequate quality.

**Table 1 sensors-20-02838-t001:** Contents of each report.

Report	Contents
1. (QL = 0)	The retinal image does not contain the OD. The patient closed eye at the moment of image capture. The distance between patient’s eye and equipment is not adequate. The patient suffers some uncommon eye disease
2. (QL = 1)	The retinal image is blurred due to one of the following reasons (1) Probable movement of the patient at the moment of image capture (2) The equipment requires some adjustment (focus, illumination). (3) The patient’s crystal is opaque
3. (QL = 2)	Some part of the ROI is obstructed by artefacts such as eyelashes and/or eyelid. The patient must try to open more widely his/her eye

**Table 2 sensors-20-02838-t002:** CNN architecture and hyper-parameters used in each operation of the proposed system.

CNN Architecture, Optimizer, Hypermeters for Training CNNs				
Operations	OD Detection	Analysis OD Region	Artefact Detection	Vessel Segmentation
CNN architectures	FR-CNN	Vgg16	SegNet	SegNet
Optimizer	SGDM	SGDM	ADAM	SGDM
Cost function	Cross Entropy	Cross Entropy	Cross Entropy	Cross Entropy
Training/Test	70%/30%	70%/30%	80%/20%	90%/10%
Minibatch size	1	64	8	32
Learning Rate	0.0001	0.0001	0.0005	0.0005
Momentum	0.9	0.9	-	0.9
Epochs	20	30	400	100

**Table 3 sensors-20-02838-t003:** OD detection performance comparison among three methods using our own dataset.

Method	Accuracy	Sensibility	Specificity	Running Time (sec)
FR-CNN Vgg16	0.7611	0.9887	0.2007	3.40
FR-CNN AlexNet (proposed)	**0.9254**	**0.9643**	**0.4424**	**0.55**
FR-CNN MobileNet	0.9228	0.9602	0.5288	2.5
Histogram matching [28]	0.3500	0.8571	0.0769	2950

**Table 4 sensors-20-02838-t004:** OD detection performance comparison among three methods using DRIMBD dataset [10].

Method	Accuracy	Sensibility	Specificity	Running Time (sec)
FR-CNN Vgg16	0.9491	0.9209	1	0.15
FR-CNN AlexNet (Proposed)	**0.9676**	**0.9481**	**1**	**0.05**
Histogram matching [28]	0.7750	0.9231	0.7037	52

**Table 5 sensors-20-02838-t005:** Performance comparison for the quality analysis of the OD region.

Methods	Accuracy	Sensibility	Specificity	Running Time (sec)
AlexNet	0.8212	0.8813	0.7585	0.02
**Vgg16**	**0.8612**	**0.9113**	**0.8064**	**0.03**
CPBD [29]	0.5737	0.5875	0.5145	0.07
Blur Metric [30]	0.5910	0.6161	0.5361	0.03
CPBD [29] + Blur Metric [30]	0.6131	0.6489	0.5574	0.06

**Table 6 sensors-20-02838-t006:** Performance comparison for the vessel segmentation using own dataset.

Method	Accuracy	Sensibility	Specificity	Running Time (sec)
SegNet_Vgg16 (proposed)	**0.9784**	**0.7169**	**0.9816**	**7.18**
SegNet_MobileNet	0.9648	0.4135	0.9934	4.14

**Table 7 sensors-20-02838-t007:** Performance comparison for the vessel segmentation using DRIVE.

Method	Accuracy	Sensibility	Specificity
SegNet_Vgg16 (proposed)	**0.937**	**0.842**	**0.938**
Palomera-Perez et al. [32]	0.925	0.640	0.967
Vlachos et al. [33]	0.929	0.749	0.955
Espona et al. [34]	0.935	0.743	0.962
Soomro et al. [35]	0.948	0.739	0.956

**Table 8 sensors-20-02838-t008:** Parameters used to generate distortions of the retinal images for agreement grade evaluation.

Distortion Effect.	Factors	Range
Blurriness	Standard deviation σ in 2D Gaussian Filter	[0, 31.5] with step 3.5
Luminance	Change of luminance ±δ	[0, 25] with step 5.0
Contrast	Standard deviation in resultant image	[0.6, 2.8] with step 0.2

**Table 9 sensors-20-02838-t009:** Agreement grade indicated by Cohen’s kappa index.

	E1 and E2	$Av\left(\begin{array}{c} \kappa \left(E_{i},G_{j}\right),\\ i=1,2,\;j=1\ldots 5 \end{array}\right)$	$Av\left(\kappa \left(E_{i},P\right),i=1,2\right)$
Cohen’s kappa	0.7199	0.5404	**0.6146**

**Table 10 sensors-20-02838-t010:** Elapse time from image acquisition to each quality level.

Quality Level (QL)	Elapse Time from Image Acquisition (sec)
QL = 0	2.49
QL = 1	2.93
QL = 2	11.3

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
