# Peer review of "A Teleophthalmology Support System Based on the Visibility of Retinal Elements Using the CNNs"

_sensors, 2020, doi:10.3390/s20102838_

Round 1

Reviewer 1 Report

The writing style needs substantial improvement to make it more structural. Thourough checkproof of English is needed. Number of grammatical errors presents (there are too many errors to list them all here). Many, overly long sentences making the reading difficult. A number of typos, also.

General questions :

It is not clear why it is important to segment the different organs (OD, vessels, … ).

Correct position of the view should also be checked. The ophtalmologists use to take nasal, central or lateral views.

Some architecture choices seem weird and should be justified. For example, regarding the OD. You train a VGG16 from scratch for detection of OD but use transfer learning for the quality estimation. Why so? Also, a VGG16 is a huge model, my feeling is perhaps even too big for that task. In the context of portable ophtalmoscope a smaller model (such as SSD MobileNet) would be more pertinent for object detection. The SSD MobileNet has ~1M parameters whereas the VGG16 more than 15M.

You would obtain a shorter processing time with smaller networks. Which indeed also seems to be one of your concerns. A short running time is important for the patient to avoid taking supplementary photos if one is already of sufficient quality. You certainly do not want to let the patient wait too long after each photograph.

How the best overall quality image is chosen based on several partial quality criteria ? According to Fig. 2, to my understanding, it is either a sum or sequence of several, binary QL checks/failures. If all QL are OK, then the image is accepted and acquisition stops. If not, at most 5 photographs are taken. How do you choose the best one ? How do you substantiate the choice ?

Overall appreciation : this works consists of a few applications of various CNN to perform a specific task.

Evaluation ? I have doubts regarding the evaluation. You claim using sensitivity, specificity and accuracy. But what measures do you use for these various tasks, such as OD, vessel segmentation, quality estimation ? The different nature of these tasks will probably require using different measures : DICE, jaccard, … etc.

Regarding the comparison with existing, the sensitivity of reference [25] in Table 3 seems really poor, far below what’s reported in their paper and on public datasets. Did you implement it yourself ? Are you fair with the implementation ? The reported timing also is weird.

The comparison with results in the literature should be extended. The literature is abundant on this subject.

Precise what is meant by « expert » and « general practitioner ».

A suggestion : could you make the images brighter and larger for printing purposes ?

Author Response

Your comment:

The writing style needs substantial improvement to make it more structural. Thourough checkproof of English is needed. Number of grammatical errors presents (there are too many errors to list them all here). Many, overly long sentences making the reading difficult. A number of typos, also.

Our response:

Thank you for reviewing our paper and we apologize for the many English errors in the paper. We carefully revised the paper and corrected grammatical errors and typos. We hope that the revised version has been improved according to the Journal requirements.

  1. It is not clear why it is important to segment the different organs (OD, vessels, … ).

Our response:

Thank you for your questions. Regarding your question, we provide the following response, from the ophthalmologist points of view. In the revised version, we added a brief description to clarify this issue.

The retina is the only organ that can be assessed through a camera without the need of invasive procedures. In order to have a comprehensive evaluation of it, ophthalmologists need to delimit the most important landmarks for the retina, which are the vessels and the optic nerve which can be observed inside of the OD. The good visibility of those elements is absolutely important to detect or diagnose a big array of eye diseases. For example, the good visibility of the optic nerve is fundamental in detecting glaucoma or other neuro-ophthalmologic or vascular diseases. One of the most common ocular diseases is diabetic retinopathy, a disease that starts on the retina vessels highlighting the importance of segmenting them as well. Finally, the segmentation of the vessels and the optic nerve, allow us to delineate retina areas such as the macula which is the area between the temporal vascular arcades and the most important part of the retina where the reading vision is located and where multiple eye diseases start. We added one sentence (lines 91-94) related to it in the revised version.

  1. Correct position of the view should also be checked. The ophthalmologists use to take nasal, central or lateral view.

Our response:

Thank you for your observation. If we use a common fundus camera with the FOV of 45° to acquire the retinal images, it is required to take several images (nasal, central and lateral views) in order to have an image composition of the retina. However, it is not necessary in our research because we used scanning laser ophthalmoscopes (SLO) as our principal device to capture the images. This SLO let us get ultra-widefield retina images; this means that with a good quality image we can obtain information of around 200° of the retina in contrast with other models that get 30° to 45°. This characteristic let us dispense the need of having the patient to turn the head at different directions in order to obtain an image-composition. In the teleophthalmology environment the central 200° are the most valuable ones, if a pathology in the periphery is suspected, the patient should always go to a specialist. In the revised version, the descriptions of the SLO mentioned above were added to clarify this issue.  We added one sentence (lines 47-48) in the revised version.

  1. Some architecture choices seem weird and should be justified. For example, regarding the OD. You train a VGG16 from scratch for detection of OD but use transfer learning for the quality estimation. Why so? Also, a VGG16 is a huge model, my feeling is perhaps even too big for that task. In the context of portable ophtalmoscope a smaller model (such as SSD MobileNet) would be more pertinent for object detection. The SSD MobileNet has ~1M parameters whereas the VGG16 more than 15M.

Our response:

Thank you for your comment about the selection of the CNN models. For the selection, we referred the following paper:

2017. Huang et al. “Speed/accuracy trade-offs for modern convolutional object detectors”, IEEE Conf. on Computer Vision and Pattern Recognition, 2017.

In this paper, authors show that the Faster-RCNN with VGG16 provides a better accuracy compared with the SSD MobileNet, when the interest object is small. In our case, the retinal image has 3900 x 3072 pixels, while the size of the OD is approximately 165x165 pixels, which means that the OD, our interest object, is very small occupying only 0.2% in the whole image. So, we selected the Faster-RCNN with the VGG16 according to the suggestion of Huang’s paper.  In the revised version, we added the justification of choice of the CNN in lines 197-203.

About your comment “You train a VGG16 from scratch for detection of OD but use transfer learning for the quality estimation.”, we used transfer learning in both cases (OD detection and quality assessment). In the revised version, we added line 221.

About your comment “Also, a VGG16 is a huge model, my feeling is perhaps even too big for that task. In the context of portable ophtalmoscope a smaller model (such as SSD MobileNet) would be more pertinent for object detection. The SSD MobileNet has ~1M parameters whereas the VGG16 more than 15M.”

We agree to your comment for portable ophthalmoscope. However, we do not consider the portable ophthalmoscope. The proposed teleophthalmology support system is considered as a part of our teleophthalmology project, in which the computation will be carried out in the cloud environment. In this sense, the memory space is not primary requirement, although we consider that the MobileNet can be a good alternative. The SLO that we used to capture the images is not portable, such that the SLO must be in a fixed position, even if the proposed system is used in a mobile clinic.  In the revised version, we added lines 560-563 to describe briefly above-mentioned issue.

In the previous version of the paper, we did not mention about our future project and principal reasons of the CNN model selection. In the revised version, we justified the selection of the CNN models.

  1. You would obtain a shorter processing time with smaller networks. Which indeed also seems to be one of your concerns. A short running time is important for the patient to avoid taking supplementary photos if one is already of sufficient quality. You certainly do not want to let the patient wait too long after each photograph.

Our response:

Thank you for your useful comment. We agreed that the execution time must be reasonably short. The elapse time that the proposed system determines that one retinal image is high quality, is approximately 11.3 sec from taking photograph. This time is average obtained using 10 high quality images. We hope that the execution time can be reduced further in the cloud environment. In the revised version of the paper, we added the average elapse time in the different quality level in table 10 (line 516) and brief description in the lines 515-519.

  1. How the best overall quality image is chosen based on several partial quality criteria ? According to Fig. 2, to my understanding, it is either a sum or sequence of several, binary QL checks/failures. If all QL are OK, then the image is accepted and acquisition stops. If not, at most 5 photographs are taken. How do you choose the best one ? How do you substantiate the choice ?

Our response:

The retinal image must be passed three binary checks: the OD detection as the first check, the OD quality as the second check, and the artefact presence in the ROI as the third check. The retinal image that pass through these three binary checks is considered as good-quality image. As you mentioned: “If all QL are OK, then the image is accepted and acquisition stops”, thus in the best case only one photograph is taken. On the other hand, in the worst case, where 5 photographs must be taken, the photograph with a larger QL value is selected as better-quality image among them. If there are several photographs with the same quality level (QL), the last one is selected as the better one. Thank you for your comment, in the revised version of the paper, we explain better the process of the proposed system (lines 186-189)

  1. Evaluation : I have doubts regarding the evaluation. You claim using sensitivity, specificity and accuracy. But what measures do you use for these various tasks, such as OD, vessel segmentation, quality estimation ? The different nature of these tasks will probably require using different measures : DICE, jaccard, … etc.

Our response:

Thank you for your valuable suggestion. We introduced the DICE and jaccard metrics to evaluate our vessel segmentation method in the revised version. In the table 6, we added the values of these two metrics. We revised several vessel segmentation methods in the literature; however, we cannot find any method that uses these two metrics (DICE and jaccard) or one of them for the performance evaluation. These metrics are commonly used for segmentation of the eye-disease, such as hard exudates.  We found a recent paper, in which authors used DICE loss function instead of a common loss function, such as cross-entropy, in the U-net CNN, however they reported accuracy, sensibility and specificity of their performance. In the revised version, we added lines 432-434 for values of the DICE and Jaccard metrics, and the performance of “U-net with DICE loss-function” in the table 7 for comparison purpose.

  1. Regarding the comparison with existing, the sensitivity of reference [25] in Table 3 seems really poor, far below what’s reported in their paper and on public datasets. Did you implement it yourself ? Are you fair with the implementation? The reported timing also is weird.

Our response:

The result of the ref. [25] in Table 3 was obtained by us, because any OD detection method for images taken by the ultrawide-FOV SLO has not been reported in the literature. We evaluated the performance of the OD detection using our own database and DRIMBD, which is only public database for retinal image quality. Both databases contain many low-quality images. In the method [25], a template of histogram of the OD region is constructed, and histograms of all blocks are compared with the template to select one block with highest similarity as the OD. This method performs well if almost all images of the dataset are good quality, however the performance is drop down when dataset contains several low-quality images, detecting no OD region as the OD. Concerning the running time of [25], the method computes the histogram of each block and the number of blocks is proportional to the number of pixels, then the running time is also proportional to the size of the image. The size of images of our own dataset (3900 x 3072) is much larger than the public datasets (for example, 565 x 584 in DRIVE). We consider that this is reason of the excessive running time of this method for our own images. In the revised version, we added our analysis in lines 374-378.

  1. The comparison with results in the literature should be extended. The literature is abundant on this subject.

Response :

Thank you for your suggestion. We agree that there are many retinal quality assessment methods in the literature. However, we cannot find any system in the literature with similar objective of the proposed system, doing it is difficult to show a meaningful comparison for the reasons mentioned below:

The principal contributions of the proposed teleophthalmology support system are:

  • Provide enough information to the operator of the SLO if the taken retinal image is low-quality for the reliable diagnosis. In this way, the operator can take a better-quality image to transmit to the central hospital.
  • The central hospital receives not only a best-quality images, but also receives several additional images, such as the OD image, binary vessel image and the image of the ROI, together with the report generated by the system. Using all information about the retinal image, ophthalmologist can do a reliable diagnostic under the absence of the patient.

Even in the literature have been reported several efficient system for image quality assessment, any of them provides to the ophthalmologist all information required for a reliable diagnostic under absence of the patient.

       In the revised version, we added some descriptions related to this issue in lines 91-94 and lines 560-563.

  1. Precise what is meant by « expert » and « general practitioner »

Our response:

We used the term “Expert” as ophthalmologist who studied more than 3 years in eye hospitals after graduating from the medical school, while we used the term “general practitioner” as a physician who graduated from the medical school, but has not finished yet the ophthalmology specialty. The experts evaluate several retinal images as his/her daily works. In the revised version, we specify these terms in lines 461-464. Thank you for your comment.

A suggestion : could you make the images brighter and larger for printing purposes ?

Our response:

Thank you for your suggestion. Following your advice, we improved quality of figures 2, 3 and 10 for the printing purposes in the revised version.

Reviewer 2 Report

This manuscript presents a system to supervise and help to improve image acquisition in teleophthalmology. Also the aim is to provide the segmentation and analysis of the principal anatomical elements of the retina. The novelty of the paper is limited because all convolutional neural networks used in the paper are well-known. The optical disc detection, quality assessment and vessel segmentation is also very common in the literature. The main contribution is that, in case of low-quality image acquisition, the system gives the possible reason causing it, so the operator can fix the problem with a new shot before sending the report to the hospital.

The paper is well structured but the English must be deeply improved. There are many mistakes regarding grammar/writing style. I think the manuscript must go under extensive reviews. From the point of view of the way the manuscript is currently presented, it is unacceptable.

From the point of view of the scientific soundness of the manuscript, it may be acceptable. The main weakness is the lack of originality in the algorithms used. Even the more specific Fast Region-Based Convolutional Neural Network applied to detect the optic disc is already presented in the paper

  1. Sadhukhan, G. K. Ghorai, S. Maiti, G. Sarkar and A. K. Dhara, "Optic Disc Localization in Retinal Fundus Images using Faster R-CNN," 2018 Fifth International Conference on Emerging Applications of Information Technology (EAIT), Kolkata, 2018, pp. 1-4. This paper is not mentioned in the manuscript.

The strength of the paper is the high number of images used to develop and train the system. Also the accuracy achieved. Regarding to the first strength, it would be very interesting, and a great contribution, if the dataset were made available to the scientific community. There are several eye and fundus image datasets like DRIMBD, MESSIDOR, etc., but a big one with Optos SLO with manually optic disc localization annotated would be very appreciated. Regarding to the second strength, it would be convenient that the system can be tested online, by means of a webpage o similar, where a random image of the dataset can be tried or third-party images can be uploaded and processed.

Additional corrections and suggestions are below.

line 203:

To generate training dataset, the bounding box is manually labeled in each training 
image of the dataset, as shown in Figure ...

What is the size of the bounding-box?

line 200:

If Pr(??) > ?h?? then the detected bounding-box is considered as OD region.

Are all 12880 images with the optic disc localization manually labelled? If so, it would be very appreciated if that such big effort were made publically available, as mentioned above.

line 236:

The link does not reach the information mentioned. There, there are a lot of committees.

https://www.gov.uk/government/groups/uk-national-screening- committee-uk-nsc

It is needed a more precise link.

line 246:

Confronting formula (1)-(8) to Figure 7, it seems that X in formulae corresponds to the vertical dimension and Y to the horizontal one. It's not common. I think this way is chosen, it must be said in the text and/or explicitly shown in the figure.

line 284:

It is not enough clear how patches are obtained from the images.

Is superpixel technique applied to whole image or to the ROI. Is it used the full resolution o the downsampled 224x224x3 version? Is downsampling the last step, applied after superpixel, and ROI cropping?

line 367:

propor?

line 372:

porposed -- proposed

line 374:

leraning -- learning

line 402:

What is the reason to try vgg16 and MobileNet for vessel segmentation instead of vgg16 and Alexnet as it was used in obstruction analysis?

line 433:

realible -- reliable

line: 436 the sentence is no completed

In this experiment, two expert ophthalmologists who are working in the retinal department of the eye hospital and five general practitioners who are working in the same eye hospital as volunteers social service.

line 438:

eveluate -- evaluate

line 441:

usefull -- useful

line 442:

distored -- distorted

line 447, Table 8

desviation -- deviation

line 482:

I think Figure 16 is not needed. More o less its contribution can be found in Figure 17.

line 488: Lases -- Laser

input retinal image taken by a non-mydriatic ophthalmoscope, such as Scanning Lases Ophthalmoscopes (SLO)

line 504

according as -- according to

line 505

We evaluated performance of each process 
 --

We evaluated the performance of each process 

Author Response

  1. This manuscript presents a system to supervise and help to improve image acquisition in teleophthalmology. Also the aim is to provide the segmentation and analysis of the principal anatomical elements of the retina. The novelty of the paper is limited because all convolutional neural networks used in the paper are well-known. The optical disc detection, quality assessment and vessel segmentation is also very common in the literature. The main contribution is that, in case of low-quality image acquisition, the system gives the possible reason causing it, so the operator can fix the problem with a new shot before sending the report to the hospital.

Response: Thank you for summarizing our paper and the principal objective of our research.

  1. The paper is well structured but the English must be deeply improved. There are many mistakes regarding grammar/writing style. I think the manuscript must go under extensive reviews. From the point of view of the way the manuscript is currently presented, it is unacceptable.

Response:  We apologize for many English errors and typos in our paper. We revised our paper carefully. In the revised version of the paper, we corrected all typos and grammatical errors.

  1. From the point of view of the scientific soundness of the manuscript, it may be acceptable. The main weakness is the lack of originality in the algorithms used. Even the more specific Fast Region-Based Convolutional Neural Network applied to detect the optic disc is already presented in the paper

Sadhukhan, G. K. Ghorai, S. Maiti, G. Sarkar and A. K. Dhara, "Optic Disc Localization in Retinal Fundus Images using Faster R-CNN," 2018 Fifth International Conference on Emerging Applications of Information Technology (EAIT), Kolkata, 2018, pp. 1-4. This paper is not mentioned in the manuscript.

Response:  We agree with your opinion, the CNNs that we used in the proposed system were proposed previously for other objectives. The OD localization method proposed by Sadnukhan et al. used the CIFAR-10 network architecture in the feature extraction part of the Faster R-CNN. The input image size of the CIFRA-10 CNN must be 32 x 32.  So, this architecture is suitable for relatively small-size interest object (the OD region), because after resizing the OD region in 32x32 pixels to be input of the CIFRA-10 CNN, the OD keep its characteristics. However, this architecture does not perform well for the large-size images (for example 3900 x 3072 pixels as the SLO’s images), because in the SLO image, the OD region is approximately 168 x 168 pixels, then the OD must be resized and  after resizing in 32x32 the relevant characteristic the OD may be lost .  

  1. The strength of the paper is the high number of images used to develop and train the system. Also the accuracy achieved. Regarding to the first strength, it would be very interesting, and a great contribution, if the dataset were made available to the scientific community. There are several eye and fundus image datasets like DRIMBD, MESSIDOR, etc., but a big one with Optos SLO with manually optic disc localization annotated would be very appreciated. Regarding to the second strength, it would be convenient that the system can be tested online, by means of a webpage o similar, where a random image of the dataset can be tried, or third-party images can be uploaded and processed.

Response:   Thank you for your useful comments. Concerning the dataset makes available to the scientific community, we totally agree to your proposal.  We will organize all images with Ground Truth (GT) and, we will star a prescribed process to public our dataset.  Concerning the proposed system can be tested online, we have plan to install the proposed system in the cloud computing environment to be tested freely by scientific community.

Additional corrections and suggestions are below.

line 203: To generate training dataset, the bounding box is manually labeled in each training image of the dataset, as shown in Figure ...

What is the size of the bounding-box?

Response:  The size of the bounding-box varies depending on the image. The mean height and width are 167.85 and 169.91 pixels, respectively. The variances of height and width are 23.94 and 26.09 pixels, respectively. In the revised version, we added size of bounding-box in lines 214-218.

line 200: If Pr(??) > ?h?? then the detected bounding-box is considered as OD region.

Are all 12880 images with the optic disc localization manually labelled? If so, it would be very appreciated if that such big effort were made publically available, as mentioned above.

Response:   Our original database size is 1288. We applied data augmentation techniques to each image to obtain in total 12880 images.  After the data augmentation, we manually delimit the bounding box for each of 12880 images using Matlab Labeling tool.  In the revised version we mentioned number of the original image in lines 136 and 216.

line 236: The link does not reach the information mentioned. There, there are a lot of committees.https://www.gov.uk/government/groups/uk-national-screening- committee-uk-nsc. It is needed a more precise link.

Response:   We apologize for this imprecision. The title of the document is “UK NATIONAL SCREENING COMMITTEE Essential Elements in Developing a Diabetic Retinopathy Screening Programme” and the link is as follows.

https://bulger.co.uk/dacorumhealth/daccom/PDF%20Documents/Diabetic%20Retinopathy%20Screening%20(Workbook%20R4.1%202Aug07).pdf

The pages 76-78 of the document mentions about the visibility of the region including macula, DO and vessels for the DR screening. In the revised version of the paper, we corrected the reference [24].

line 246: Confronting formula (1)-(8) to Figure 7, it seems that X in formulae corresponds to the vertical dimension and Y to the horizontal one. It's not common. I think this way is chosen, it must be said in the text and/or explicitly shown in the figure.

Response:   Thank you for your suggestion. To make easy to understand, in the revised version we modified equations (1)-(8) and Fig. 7 according to the common way: X is column index and Y is row index.

line 284: It is not enough clear how patches are obtained from the images.

Response:  We obtained randomly 100 patches from each one of 10 images with artefacts (eyelashes and eyelid). Then data augmentation technique is applied to these 100 patches to increase artificially to 500 patches per image. In total, we generated 5000 patches to train the SegNet.  The 80% of the total 5000 patches are used in training process and the rest 20% are used for testing. In the revised version of the paper, we modified the description to clarify how to obtain the patches in lines 296-299.

Is superpixel technique applied to whole image or to the ROI. Is it used the full resolution or the downsampled 224x224x3 version? Is downsampling the last step, applied after superpixel, and ROI cropping?

Response: Yes, we applied super-pixel technique to whole image with the full resolution. We set the number of super-pixels to 1000 for the whole image. The objective of the use of super-pixel is the effective construction of the Ground Truth of artefact pixels, such as eyelash and eyelid. The super-pixel technique allowed us generating Ground Truth (GT) of artefact (eyelash and eyelid) easier and more effective. Once binary GT images are constructed, they are used to train the SegNet CNN. As mentioned above, the same data augmentation technique is applied to the corresponding GT patches to generate 5000 GT patches. In the revised version of the paper, we amplified the description to clarify the use of super-pixel technique in lines 273-275 and lines 280-281.

line 367: propor?

line 372: porposed -- proposed

line 374: leraning – learning

Response: We apologize for these errors. We corrected these errors and carefully reviewed to avoid typos.

line 402: What is the reason to try vgg16 and MobileNet for vessel segmentation instead of vgg16 and Alexnet as it was used in obstruction analysis?

Response:   We proved SegNet with vgg16 and MobileNet for vessel segmentation task, and SegNet with vgg16 for artefacts (eyelashes and eyelid) detection. Because the SegNet with vgg16 provides a more precise segmentation. We reviewed the redaction of our paper in order to clarify that Alexnet is not used in these tasks.

line 433: realible – reliable

Response: We apologize for this error. We corrected the error and revised carefully the paper.

line: 436 the sentence is no completed

In this experiment, two expert ophthalmologists who are working in the retinal department of the eye hospital more than three years and five general practitioners who are working in the same eye hospital as volunteers social service participated in the evaluation process.

Response: We apologize for the incomplete sentence. We corrected this sentence in lines 458-461 in the revised version.

line 438: eveluate -- evaluate

line 441: usefull -- useful 

line 442: distored -- distorted

line 447, Table 8 desviation – deviation

line 482: I think Figure 16 is not needed. More o less its contribution can be found in Figure 17.

Response: Thank you for your suggestion. We eliminated Fig. 16 from the revised version of the paper.

line 488: Lases -- Laser

input retinal image taken by a non-mydriatic ophthalmoscope, such as Scanning Lases Ophthalmoscopes (SLO)

line 504 according as -- according to 

line 505 We evaluated performance of each process 
 --

We evaluated the performance of each process

Response:  We apologize for this error.  We corrected the errors and carefully revised the revised version of the paper.

Round 2

Reviewer 2 Report

Everything has been reviewed and it is fine, but the dataset with the ground-truth has not yet been published nor the website with the system.

Author Response

Response to the reviewer

Thank you for reviewing our paper.

Point 1: Everything has been reviewed and it is fine, but the dataset with the ground-truth has not yet been published nor the website with the system.

Response:  Regarding to your question, the dataset and the system will be published after it be registered in the Mexican intellectual property office. Unfortunately, due to the pandemic, all Mexican government offices are closed. Once the registration is obtained, the database will be available for the research community without cost.